# Fluorescence changes in carbon nanotube sensors correlate with THz absorption of hydration

Sanjana S. Nalige[1,5], Phillip Galonska[1,5], Payam Kelich[2], Linda Sistemich[1], Christian Herrmann[3], Lela Vukovic[2], Sebastian Kruss[1,4]✉ & Martina Havenith[1]✉

Single wall carbon nanotubes (SWCNTs) functionalized with (bio-)polymers such as DNA are soluble in water and sense analytes by analyte-specific changes of their intrinsic fluorescence. Such SWCNT-based (bio-)sensors translate the binding of a molecule (molecular recognition) into a measurable optical signal. This signal transduction is crucial for all types of molecular sensors to achieve high sensitivities. Although there is an increasing number of SWCNT-based sensors, there is yet no molecular understanding of the observed changes in the SWCNT's fluorescence. Here, we report THz experiments that map changes in the local hydration of the solvated SWCNT upon binding of analytes such as the neurotransmitter dopamine or the vitamin riboflavin. The THz amplitude signal serves as a measure of the coupling of charge fluctuations in the SWCNTs to the charge density fluctuations in the hydration layer. We find a linear (inverse) correlation between changes in THz amplitude and the intensity of the change in fluorescence induced by the analytes. Simulations show that the organic corona shapes the local water, which determines the exciton dynamics. Thus, THz signals are a quantitative predictor for signal transduction strength and can be used as a guiding chemical design principle for optimizing fluorescent biosensors.

Single wall carbon nanotubes (SWCNTs) are one-dimensional nano-materials with rich optoelectronic properties and surface chemistry[1]. They can be considered as rolled-up graphene sheets and the roll-up angle defined by the chirality (n,m) determines their photo physics[2]. Upon excitation of (semiconducting) SWCNTs by light, electrons are elevated into the conduction band, leaving a positively charged hole in the valence band. Because of the one-dimensionality of SWCNTs, these electron-hole pairs have high binding energies and diffuse as excitons along the SWCNT axis[2]. Excitons are larger than the SWCNT's diameter[3]

and therefore highly sensitive to the SWCNT's local environment[4,5]. Upon radiative re-combination of the exciton, SWCNTs emit fluorescence in the near-infrared (NIR) tissue transparency window (870 nm–2400 nm). This property has fostered the use of biofunctionalized SWCNTs in the NIR with an increasing number of applications in (bio)photonics[6,7].

Molecular sensing is becoming increasingly important in biomedical research, healthcare, and agriculture[8,9]. The use of SWCNTs as biosensors in aqueous environments requires their hydrophobic

[1]Department of Physical Chemistry II, Ruhr University Bochum, Bochum, Germany. [2]Department of Chemistry and Biochemistry, University of Texas at El Paso, El Paso, TX, USA. [3]Department of Physical Chemistry I, Ruhr University Bochum, Bochum, Germany. [4]Fraunhofer Institute for Microelectronic Circuits and Systems, Duisburg, Germany. [5]These authors contributed equally: Sanjana S. Nalige, Phillip Galonska. ✉e-mail: sebastian.kruss@rub.de; martina.havenith@rub.de

surface to be functionalized by amphiphilic modules, such as biopolymers or surfactants. One commonly used biopolymer is single-stranded DNA (ssDNA). In addition to solubilizing the SWCNT[10], this organic phase shields the exciton from the solvent, which is believed to enhance the fluorescence quantum yield of SWCNTs[11].

Upon functionalization, the nucleobases of DNA adsorb on the SWCNTs' surface by $\pi$-stacking and hydrophobic interactions, with the hydrophilic phosphate backbone of the DNA pointing towards the solvent[12]. The assembly of DNA and SWCNTs can interact with (bio-) molecules such as catecholamines[13] and different DNA- sequences cause different changes in the SWCNT fluorescence. Since the discovery of the sensing abilities of DNA functionalized SWCNTs, they found wide applications as (bio)sensors[14] from neurotransmitter imaging[13], and plant stress detection to cancer diagnostics[15–17]. However, the mechanism of fluorescence modulation is still not understood, which so far prevents further advances.

Previously, SWCNT-fluorescence changes have been explained in terms of Solvatochromism i.e., exciton-interactions with the solvent[5,14,18,19]. Solvatochromism relies on macroscopic bulk properties, such as dielectricity. However, the bulk dielectric constant of an aqueous solution does not change significantly upon the addition of μM or even lower concentrations of analytes. Thus, they cannot explain the experimentally observed large changes in SWCNT fluorescence of up to a few hundred percent[11,20].

In recent theoretical studies by the groups of Bocquet[21] and Cox[22] the concept of quantum friction induced by the coupling of charge fluctuations in the liquid to electronic excitations in the solid at water-carbon interfaces has been introduced. They investigated theoretically the non-adiabatic coupling between water Debye collective modes with a thermally excited plasmon specific to graphite. Recently there has also been experimental evidence by the group of Kavokine of this quantum contribution by probing the coupling between graphene's surface plasmon mode with water libration modes[23]. The overlap between the two spectra is most significant below 20 THz or 700 cm⁻¹. While the vibrational density of states of the excited plasmon increases with frequency, the spectrum of bulk water in the THz range is characterized by two pronounced maxima at approximately 7 and 20 THz, which could be attributed to the intermolecular hydrogen bond stretch centered at 6–7 THz, i.e., 200 cm⁻¹, and the hindered rotation of a single water molecule in hydrogen bond network, referred to as libration with a peak absorption between 10–23 THZ, i.e. 350 and 750 cm⁻¹ [24].

In the bulk, cations, and anions have characteristic rattling modes, which allow us to deduce the size of their hydration shell[25] as well as observe the formation of ion pairs using THz spectroscopy. The correlations in the solute–solvent dynamics, induced also by charge fluctuations reach beyond the first hydration shell[26] and can be probed by THz spectroscopy. Furthermore, sensitive THz absorption studies allow to probe the hydration shell to map distinct water populations in the hydration shell, such as hydrophobic hydration water, and hydrophilic hydration water[27,28].

In the present paper, we present the result of a systematic investigation of how analytes change the local solvation of a nanomaterial-based biosensor and its fluorescence. We investigate the coupling between functionalized SWCNTs and the hydration water in the low-frequency range, which can be sensitively probed by THz absorption spectroscopy[26,28,29]. We provide experimental evidence for an inverse correlation between the coupling of electronic fluctuations in the SWCNTs to the charge density fluctuations in the hydration shell, as probed by changes in the THz and changes in fluorescence yield upon the addition of an analyte. We propose that the energy transfer after optical excitation can take two distinct pathways: either the energy is efficiently transferred into the solvent or, if the coupling is less efficient, NIR fluorescence emission becomes more dominant.

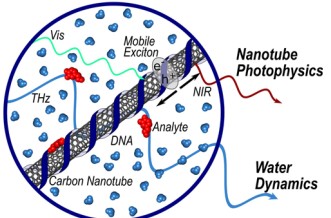

**Fig. 1 | Schematic of carbon nanotube-based molecular sensors and their spectroscopy.** Biofunctionalized carbon nanotubes change their fluorescence when they interact with analytes. THz spectroscopy probes how water around them changes. This interplay is studied to understand the sensing mechanism.

## Results
### Coupling of charge fluctuations probed in the THz range is correlated to fluorescence
We carried out THz absorption and NIR fluorescence measurements (Fig. 1) to unravel possible correlations between the charge-induced coupling of local solvation and the SWCNT's fluorescence. We colloidally stabilized the hydrophobic SWCNTs by adsorbing biopolymers such as $(GT)_{10}$ ssDNA or surfactants (deoxycholate, DOC; sodium cholate, SC, and sodium dodecyl benzene sulfonate, SDBS). This procedure creates an organic corona around the SWCNTs and makes them soluble in water and high ionic strength buffers.

Unlike in our study of solvated salts, in which rattling of the atomic charges in the water network caused a characteristic anion or cation absorption mode[25], for the solvated SWCNT, we observe a broad increase in absorption, extending over the entire frequency range. This indicates a broadband coupling in the frequency range of the translational (100 cm⁻¹), intermolecular stretch, and vibrational modes of water, as predicted by Bocquet et al.[21] and Cox et al.[22]. More quantitatively, the amplitude of $\Delta\alpha$ is a measure of the charge/carrier coupling. Based on our measurements we find that this coupling depends on the functionalization of the SWCNT, with functionalization by $(GT)_{10}$ and SDBS resulting in a more efficient coupling to the hydration shell compared to that of DOC and SC (Fig. 2a).

Upon optical excitation (fluorescence measurements) all functionalized SWCNT will create additional charge carriers, the so-called excitons, which can recombine under the emission of a photon in the NIR. In Fig. 2b, we plot the NIR fluorescence intensity for the same functionalized SWCNTs. Interestingly, we find that SWCNTs, which show a more efficient coupling to the charge fluctuations in the hydration shell, as evidenced by a higher $\Delta\alpha$, have a lower fluorescence intensity (quantum yield, Fig. 2c). This gives a hint to the underlying molecular mechanism: We propose that the energy relaxation of excitons can take place either via an efficient coupling into the solvent (as probed in the THz) or via recombination of the charge carriers (as probed by NIR fluorescence). Therefore, the spectroscopic observables of each of these alternative energy pathways will be inversely correlated.

We propose that the different spectral properties of functionalized SWCNTs can be explained by the fact that DNA is more loosely bound to SWCNTs compared to sodium deoxycholate (DOC) or sodium cholate (SC). This enables the charge density fluctuations in the water to couple more strongly with the SWCNTs for DNA or SDBS functionalized SWCNT than in the case of DOC or SC. Because of this coupling of SWCNTs to the charge density fluctuations of water, the number of charge carriers in SWCNTs that undergo recombination is reduced. This corresponds to the quenching of fluorescence and consequently, a decrease of the fluorescence quantum yield given that the absorption cross-section does not depend on the surfactant[30]. In the case of DOC or SC, the coupling between the charge fluctuations of water molecules in the solvation layer and the SWCNTs is reduced, as

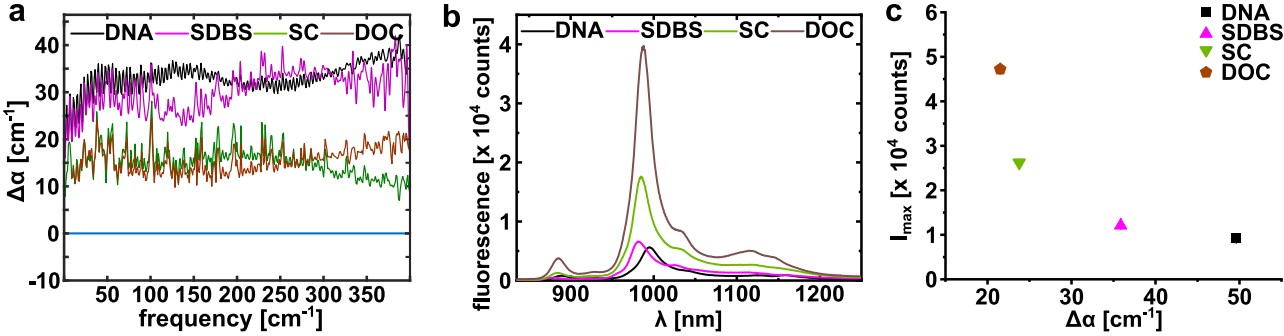

**Fig. 2 | Single wall carbon nanotube (SWCNT) functionalization inversely governs THz absorption and intrinsic fluorescence. a** THz difference spectrum, $\triangle\alpha$ (= $\alpha_{sample} - \alpha_{water}$) of (6,5)-chirality enriched semiconducting single-walled carbon nanotubes SWCNTs with different surface chemistries/functionalization. Spectra is a mean of two measurements of distinct samples carried out on the same day (**b**) Fluorescence spectra of SWCNTs with the same functionalization as shown in **a**. Spectra are mean values of three subsequent measurements. **c** Maximal fluorescence of the samples shown in **b** plotted against the difference in THz absorption. Fluorescence data are represented as mean values ± SEM ($n$ = 3).

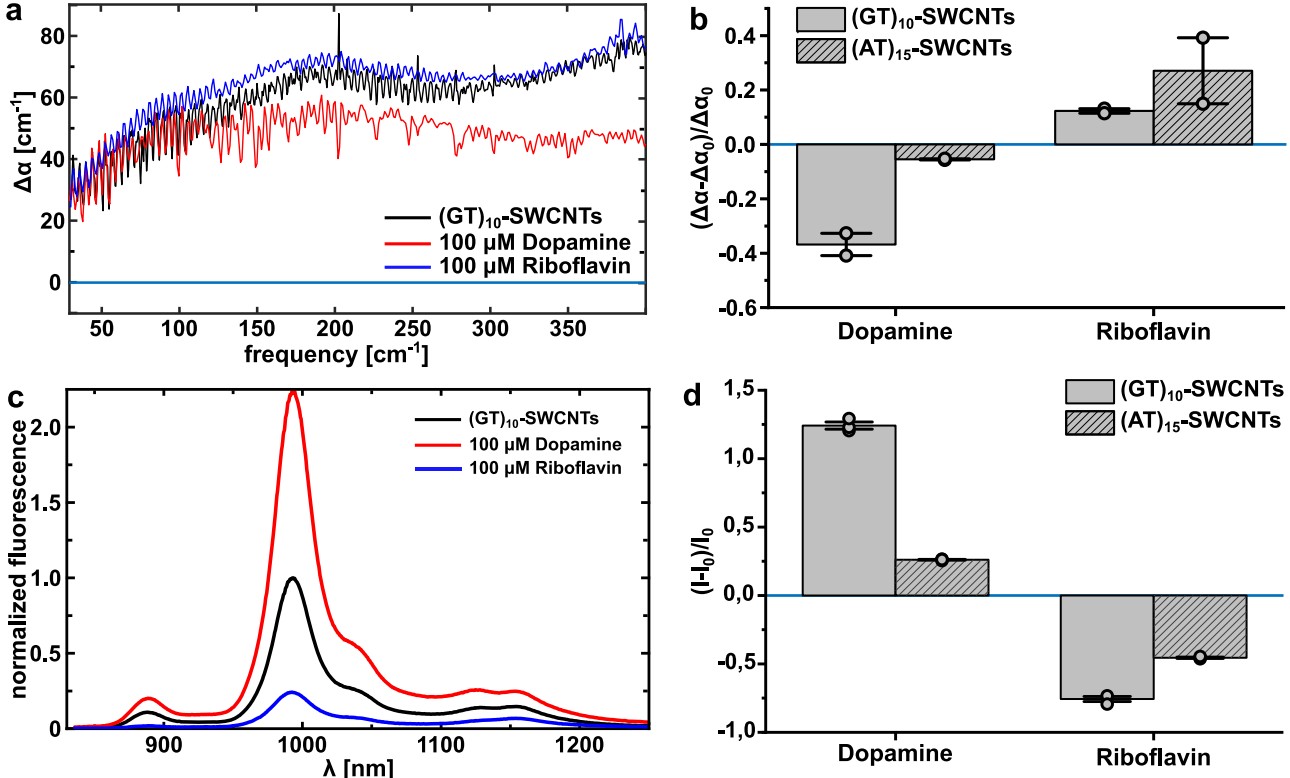

**Fig. 3 | Analyte binding affects coupling of the hydration shell and single wall carbon nanotube (SWCNT) photophysics. a** Difference spectra, $\triangle\alpha$ (= $\alpha_{sample} - \alpha_{water}$) of (GT)$_{10}$-SWCNTs, i.e., change in THz absoption upon addition of analytes (100 μM) in physiological buffer. (All spectra are corrected for water vapor lines). **b** Change in difference spectra upon addition of the analyte at $v$ = 150 cm$^{-1}$ compared to the absorption of the solvated (GT)$_{10}$- and (AT)$_{15}$-SWCNTs. Data is represented as mean values ± SEM ($n$ = 2). **c** Representative NIR-fluorescence spectra of (GT)$_{10}$-SWCNTs before and after analyte addition (100 μM) in physiological buffer. **d** Normalized fluorescence changes of (GT)$_{10}$ and (AT)$_{15}$-SWCNTs after addition of 100 μM respective analyte. Data are represented as mean values ± SEM ($n$ = 3).

confirmed by the smaller Δα (Fig. 2a) but in turn, the quantum yield for fluorescence is increased (Fig. 2b).

To validate this model, we tested (GT)$_{10}$-SWCNTs, which are highly selective NIR fluorescent biosensors for the neurotransmitter dopamine and the vitamin riboflavin. It is known that upon addition of the analytes dopamine or riboflavin, the impact on the quantum yield is opposite i.e., the fluorescence intensity increases for dopamine and decreases for riboflavin[13,31,32]. While the effect is known, the underlying mechanism has not been understood so far. In Fig. 3a, we plot Δα, i.e., the THz difference spectra upon the addition of either dopamine or riboflavin to the (GT)$_{10}$-SWCNTs concerning water. Per our working hypothesis that a decrease/increase in THz absorption goes along with an increase/decrease in fluorescence, we found that upon addition of dopamine, epinephrine, and ascorbic acid Δα decreased and for riboflavin Δα increased, compared to the solvated (GT)$_{10}$-SWCNTs (Fig. 3a and Supplementary Fig. 1). For direct comparison, we also measured changes in the fluorescence of the DNA-SWCNTs upon the addition of these analytes (Fig. 3c, and Supplementary Figs. 2, 3). All analytes were added at the same concentration (100 μM) as in the THz spectra. Additionally, we carried out the above-mentioned

experiments with $(AT)_{15}$-SWCNTs (Fig. 3b, d, Supplementary Fig. 4). The fluorescence and THz absorption responses of $(AT)_{15}$-SWCNTs followed the same trend as seen for $(GT)_{10}$-SWCNTs, but the magnitude of change in fluorescence upon the addition of analytes is smaller. Since the coupling of the electronic fluctuation of SWCNTs to charge density fluctuations in water depends on the specific SWCNT corona (Fig. 2), the alterations induced by the analytes can vary in magnitude. The chirality of the SWCNTs could play a certain role in the sensing mechanism. In our experiments we used (6, 5)-SWCNTs. It was shown before that the chirality of DNA-SWCNT-based sensors for small molecules plays a less important role than the surface chemistry (e.g., DNA sequence[33], compare Supplementary Fig. 5). Other factors such as orientation could also play a role but should be negligible in solution[34].

In the THz spectra (Fig. 3a), we observed a broad increase in absorption between $100\,cm^{-1}$ and $200\,cm^{-1}$. Remarkably, these two frequencies correspond to the expected center peak frequencies for the translational and intermolecular stretch of bulk water, respectively. A further increase in $\Delta\alpha$ is observed for frequencies $>350\,cm^{-1}$, i.e., at the onset of the librational mode. This experimental observation supports our hypothesis that $\Delta\alpha$ probes the coupling between the THz spectrum of water and the SWCNTs, which is predicted theoretically. To quantify the coupling, we plotted in Fig. 3b the change in difference spectra upon the addition of the analyte at $300\,cm^{-1}$ (for $\nu = 150\,cm^{-1}$ see Supplementary Fig. 6) compared to the absorption of the solvated $(GT)_{10}$−SWCNTs, i.e., $(\Delta\alpha$-$\Delta\alpha_0)/\Delta\alpha_0$, which yielded an inverse correlation between both quantities.

Thus, based on our experimental results, we conclude that the addition of analytes impacts the coupling between the DNA-SWCNT and the hydration water, which in turn changes the fluorescence quantum yield as can be seen by the fluorescence intensity changes (Fig. 3c, d). Note that fluorescence intensity and fluorescence quantum yield are proportional to each other because the absorption spectra of the SWCNTs are not changed by the analyte (Supplementary Fig. 7)[13].

For SWCNTs, it has been proposed that there is a strong coupling between surface plasmon modes in SWCNTs and excitons[35]. As the energy of (axial) surface plasmons of SWCNTs lies in the THz region[36] we propose that this is a dissipation pathway. The spectral overlap between excitons and THz modes of water alllows coupling. An alternative could be the large number of (dark) states within the bandgap of the SWCNT.

In Fig. 4a, we plotted the normalized change in fluorescence intensity against the normalized change in $\Delta\alpha$. We observe an inverse correlation between the change in quantum yield/NIR fluorescence and the change in $\Delta\alpha$. Whereas the same trend is seen for analytes with $(AT)_{15}$ and $(GT)_{10}$ SWCNT, the correlation factor, i.e., the value of the slope, is found to be DNA-sequence specific. The $(GT)_{10}$ functionalized SWCNT responds strongly to the binding of the analyte. Therefore, the slope, which serves as a guide to the eye is steeper than for $(AT)_{15}$ functionalized SWCNT (Fig. 4a).

This pattern further corroborates our theory of competing pathways for the relaxation of charge carriers/excitons in SWCNTs. Based on the changes the analytes create in the local environment, the coupling between water and SWCNTs changes. The fluorescence signal is higher when the coupling is lower and vice versa (Fig. 4b). To explain these differences, we carried out accompanying molecular dynamics (MD) simulations (Fig. 5).

Hence, the SWCNT corona is impacted by binding to an analyte. In general, analytes that increase fluorescence, such as dopamine, reduce the observed THz coupling efficiency, whereas quenching analytes, such as riboflavin, increase the coupling efficiency. The underlying molecular alterations of the corona are not revealed by the spectroscopical data. To understand the changes happening in the corona we carried out accompanying molecular dynamics (MD) simulations (Fig. 5 and Supplementary Fig. 8).

They underline the differences in the interactions between $(GT)_{10}$-wrapped SWCNTs and two analyte molecules, dopamine, and riboflavin. In simulations of DNA-SWCNTs with dopamine, three out of ten dopamine molecules and four out of ten dopamine molecules are directly bound to the SWCNT surface, as seen in Fig. 5a. The remaining dopamine molecules are bound to the DNA corona. In simulations of DNA-SWCNTs with riboflavin, two out of ten molecules and three out of ten riboflavin molecules directly bound to the SWCNT surface (Fig. 5b). However, riboflavin molecules were either tilted with respect to the SWCNT surface (Fig. 5b) or were associated purely with the DNA corona. Overall, riboflavin molecules exhibited a preference for binding to the DNA corona. Based on these observations, we propose two general binding modes: Mode 1 where the analyte binds directly to the SWCNT surface, and mode 2 with analytes binding to the corona. Figure 5c provides a schematic of each of these modes. In the SI (Supplementary Tables 1 and 2) we provide the details of the simulated systems and binding events in the systems characterized by the type of the binding mode, and the number of carbon atoms of the SWCNT in the vicinity of the bound ligands.

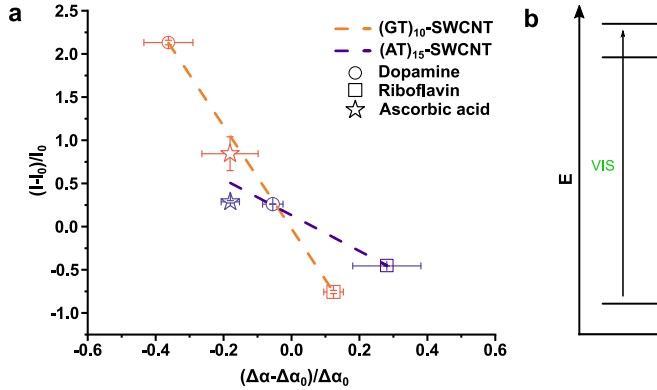

**Fig. 4 | Fluorescence quantum yield is inversely proportional to coupling between single wall carbon nanotube (SWCNT) and water as probed by $(\Delta\alpha$-$\Delta\alpha_0)/\Delta\alpha_0$. a** Plot of the relative change of $\boldsymbol{\alpha}$ (i.e., change in THz absorption) vs relative change of I (i.e., change in fluorescence intensity, which is a measure for quantum yield) upon addition of analytes dopamine, ascorbic acid, and riboflavin. In this plot, all values for $\Delta\alpha$ are taken at $150\,cm^{-1}$ wavenumbers. The data points shown include data of the SWCNTs functionalized with two distinct DNA sequences, $(AT)_{15}$ and $(GT)_{10}$. Dashed lines represent linear fits for both DNA sequences with slopes $m_{(GT)10} = -4.11557$ and $m_{(AT)15} = -2.07549$. Data are represented as mean values ($n = 3$). Error bars = SEM ($n = 3$). **b** Simplified energy state schema for excited SWCNTs suggesting an alternative non-radiative decay (NRD). GS: Ground state; $E_{11}/E_{22}$: SWCNT excited states; VIS: Visible excitation; NIR: NIR fluorescence emission. Arrows indicate the transition between states.

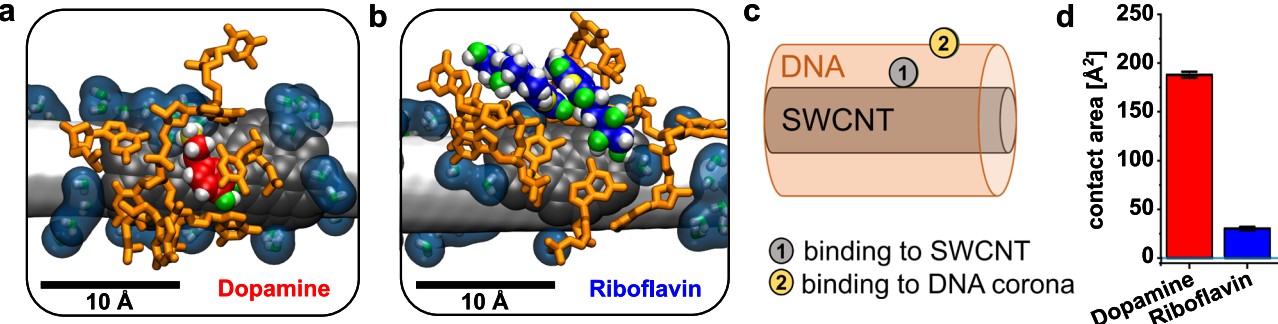

**Fig. 5 | Effects of ligand binding on the SWCNT's (single wall carbon nanotube) hydration shell probed by molecular dynamics simulations. a** Snapshot of a dopamine ligand directly contacting the SWCNT surface. Carbon, oxygen, nitrogen, and hydrogen atoms of dopamine are depicted in red, green, yellow, and white, respectively. DNA appears in orange, SWCNT in white, water in dark blue, and the dark gray region represents the calculated localized water contact area. **b** Snapshot of two riboflavin ligands in direct contact with the SWCNT surface. The color scheme mirrors panel a, with riboflavin's carbon atoms in blue. **c** Schematic of the two preferred binding modes for dopamine (SWCNT surface) and riboflavin (DNA corona). **d** Bulk contact areas between the SWCNT surface and all dopamine (DA-red) or riboflavin (RBF-blue) molecules. Data are presented as mean values ± SEM ($n$ = 30,000 configurations extracted from 3 μs MD trajectories), where averaging is performed over the trajectory time.

To assess the influence of the analyte on the bulk contact area between SWCNT and water/ions, we performed an analysis on five simulated systems, including a control system without a ligand. Two systems contained dopamine molecules, and two systems contained riboflavin molecules. While no significant differences were observed in the contact areas between SWCNT surface to water and sodium ions across all systems, the contact areas between ligands and the SWCNT surface exhibited substantial variations. Specifically, dopamine molecules demonstrated a higher contact area with the SWCNT surface compared to riboflavin (Fig. 5d). This direct binding was confirmed experimentally by isothermal titration calorimetry (ITC). Dopamine was titrated in $(GT)_{10}$-SWCNTs to a total concentration of 100 μM (Supplementary Figs. 9, 10 for riboflavin, and Supplementary Table 3). Based on these experiments we deduce an interaction stoichiometry ranging from 1:30 to 1:40 (SWCNTs: dopamine), meaning that a single SWCNT covered with ssDNA harbors on average 37 strong interaction sites for dopamine. They correspond to higher affinity states on the SWCNT surface with a larger thermodynamic footprint. The lower affinity binding sites in the DNA corona showed a smaller thermodynamic footprint and did also not saturate at the high concentrations up to 100 μM. This result verifies the two different binding modes found in the MD simulations.

The analysis of the overall contact areas between the SWCNT surface and water or sodium ions for modeled systems, including the control $(GT)_{10}$-wrapped SWCNT system (no analytes) and $(GT)_{10}$-wrapped SWCNT systems binding to dopamine or riboflavin analytes. These bulk analyzes of the systems revealed no significant overall effects of analytes on the environment of the SWCNTs. Given these results of the bulk analysis, the localized contact areas between the SWCNT surface segments and surrounding water molecules were examined next (Supplementary Fig. 11). In this approach, we tracked the number of water molecules near local regions of SWCNTs, comprised of a fixed number of SWCNT carbon atoms in the vicinity of the bound ligands. We identified local regions for all ligand binding events that last longer than 400 ns. In Supplementary Fig. 12 we plotted the change in the average contact area between water and the local SWCNT surface when analytes are binding to the regions and when they are not binding. Since the sizes of the local SWCNT regions can differ, we normalized the water contact area values by dividing each value by the maximum value observed for the contact area of water. Local SWCNT regions had less direct contact with water when the analytes bind to the local region (in binding mode 1, as for dopamine) than when they bind to the DNA corona (mode 2, as for riboflavin). In fact, the water-region contact areas were similar when the ligands

bound to the DNA corona (binding mode 2) and when the ligands left the region, indicating that ligand binding to the corona does not affect the exposure of local SWCNT regions to water. These results together with the THz results indicate that one analyte (riboflavin) affects the water structure in the DNA corona while the other analyte (dopamine) affects mainly the water structure directly on the SWCNT surface.

In the case of dopamine, we expect a bigger impact on the coupling between charge fluctuations of the water and the SWCNTs than in the case of the more loosely bound riboflavin. Therefore, in the case of dopamine, we observe a reduction in the coupling of the SWCNT and the water, as quantified by Δα, and thus an increase in quantum yield, whereas for analytes that quench SWCNT fluorescence (e.g., riboflavin) the opposite holds. Thus, local solvation changes at the SWCNT surface affect the fluorescence of SWCNTs. However, in contrast to previous common bulk solvatochromic models of SWCNTs that predict wavelength shifts if SWCNTs are exposed to more or less water, there is no overall net change in SWCNT exposure to water[5,11]. The solvatochromic models were unable to predict the fluorescence increases or decreases observed for many SWCNT-based sensors including the ones studied here[1]. The model presented here gives now a conclusive explanation for the underlying molecular mechanism.

## Discussion

Optical molecular sensors/probes are important tools for basic research and biomedical applications. However, up to date, in most cases, it is not understood how the binding of an analyte ultimately affects the optical properties of the material or fluorophore. Previously, we introduced a kinetic model to describe exciton fate in DNA-SWCNTs[37]. It could model the fluorescence changes by taking into account that analytes affect photophysical rate constants. In this picture, excitons undergo radiative decay (fluorescence) or non-radiative decay (e.g., quenching at a SWCNT end or transition to a dark state[38]).

Here, we investigate the coupling of the low-frequency DNA-SWCNT mode with the hydration water via THz absorption. We propose that the change in absorption reports on the energy transfer between exciton-coupled surface plasmons of DNA-SWCNTs and the translational and hydrogen bond stretch mode. The efficiency of this coupling is proposed to affect the magnitude of the broadband correlated solute-water mode and can be modified by the binding of analytes depending on the binding motif. Upon optical excitation, the energy can either be transferred efficiently to the solvent (via correlated DNA- SWCNT/ hydration water modes), which are proposed to be similar in the ground and excited electronic state or will finally lead to an exciton decay via fluorescence. Thus, in this picture excitons

undergo radiative decay (fluorescence) or non-radiative decay (e.g., quenching at a SWCNT end or transition to a dark state).

While it was known before, that these transition rates are affected by the presence of the analyte, the physicochemical origin remained unknown. Now, we show that coupling between the DNA-SWCNTs and the hydration shell dictates these transition rates. The exciton screens the first and higher-order solvation shells, which are perturbed by analyte binding[39]. Therefore, in the case of SWCNT-based fluorescent nanosensors, THz screening could serve as a predictor for the fluorescence yield.

Our findings support explanations in the literature[13] that propose conformational changes in the SWCNT's corona as the reason for analyte-induced fluorescence changes. Most interestingly, our work shows that conformational changes impact the coupling between the low-frequency modes of the SWCNT and the intermolecular modes of the hydration water. Facilitating or preventing this alternative non-radiative decay channel is postulated to be the underlying molecular mechanism for the impact on the radiative exciton decay via charge carrier recombination. This coupling could serve as a general design principle to develop biosensors with stronger signal transduction and can be transferred to other nanoscale sensors and systems. Changing the surface chemistry (e.g., DNA sequence) allows for tuning of coupling strength with the solvent and represents a rational chemical approach for biosensors of improved sensitivity.

## Methods
### Functionalization of SWCNTs
To disperse SWCNTs in DNA, one part 2 mg/ml of (6, 5)-enriched SWCNTs (Merck) in 1 x phosphate buffered saline buffer solution (PBS, Carl Roth) was added to two parts 2 mg/ml ssDNA (Merck) in 1 x PBS to a total volume of 1.5 ml. The mixture was placed in a tip sonicator (Fisher Scientific) and sonicated for 30 min at 60 % amplitude (72 W output power) in an ice-water bath. After sonication the sample was centrifuged at $21,000 \times g$ for 30 minutes, the supernatant was transferred to a new vial and the procedure was repeated two more times. The stock concentration was determined by absorption spectroscopy using a VIS-NIR V-770 spectrometer (Jasco). To adjust the concentration the maximum intensity and full width at half maximum of the $E_{11}$ absorption feature of the (6, 5) chirality was used[39,40]. For THz spectroscopy experiments, the SWCNT samples were diluted to a final concentration of 100 nM in PBS.

### NIR Fluorescence measurements
For excitation of SWCNTs a 561 nm Laser (Laser Quantum) was used. Spectra were measured using a Shamrock 193i spectrometer (Andor Technology Ltd.) equipped with an Andor iDus InGaAs 491 detector. Laser and spectrometer were coupled to an IX73 microscope (Olympus)[37]. The NIR-fluorescence response was measured by adding 2 μl of analyte solution to 198 μl of 1 nM SWCNT solution. Spectra were recorded with 100 mW laser excitation power, 2 sec exposure time, and 500 μm spectrometer slit width. Dopamine, riboflavin, and ascorbic acid were purchased from Merck. Triplicates were collected from distinct samples.

### Measurement of coupling between SWCNT and hydration shell
We recorded the THz response of the solvated SWCNT using a Fourier transform infrared (FTIR) spectrometer (Bruker Vertex 80 v, Billerica, MA). The samples were measured in a Bruker liquid cell with chemical vapor deposition-grown diamond windows ($500 \pm 100$ μm thickness; Diamond Materials, GmbH) and 13 μm-thick Kapton spacers for the measurements. The sample chamber was continuously purged with dry nitrogen at -2 bar and the sample cell temperature was kept at 24 °C using a chiller; the sample compartment was separated by polyethylene flaps from the rest of the evacuated spectrometer chamber. The precise layer thickness was determined by recording the

etalon fringes of the empty cell by mid-infrared absorption spectroscopy. The absorption was measured in the frequency ranges from 50 to 500 cm⁻¹ (1.5–12 THz) as an average of over 64 scans.

The resulting total absorption coefficient of a solution is determined as:

$$\alpha = -\frac{1}{d}\log\left(\frac{I}{Io}\right)$$

$$\triangle\alpha = \alpha_{\text{sample}} - \alpha_{\text{water}}$$

with 'd' being the measured layer thickness, I and $I_0$ being the transmitted intensities of the sample and background, respectively. More details of our setup and analysis can be found in SI (Supplementary Fig. 13) and ref. 41. The difference spectra are calculated to remove bulk water contributions and observe the changes in local water molecules. Changes in the extent of coupling of the plasmonic modes in the SWCNT to the charge density fluctuations in the hydration layer can be investigated using $\triangle\alpha$. Control measurements with solid SWCNTs, dried SWCNTs + DNA, buffer, buffer + analytes and time-dependent measurements can be found in SI (Supplementary Figs. 14, 15, and 16).

### Molecular Dynamics (MD) simulations
Atomistic models of ssDNA-SWCNT conjugates. A computational system of eight $(GT)_{10}$ DNA molecules adsorbed on a 12-nm long segment of (6, 5) SWNT in 0.155 M NaCl aqueous solution was prepared and examined in MD simulations. The prepared system was examined by itself (control system) and when binding to either ten dopamine molecules or ten riboflavin molecules. Two independent simulations of each studied system were carried out. Further details of model preparation, MD simulations, and analyses of these systems can be found in SI.

### Reporting summary
Further information on research design is available in the Nature Portfolio Reporting Summary linked to this article.

## Data availability
All the data generated in this study has been deposited in our open-access repository: https://doi.org/10.17877/RESOLV-2024-lxymp01c.

## Code availability
The code to generate the data presented in Supplementary Fig. 5 is available in our open-access repository: https://doi.org/10.17877/RESOLV-2024-ly8p38e6. The code to generate the presented MD data is available in an open-access repository https://zenodo.org/records/12667707.

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

## Acknowledgements

This work was funded by the Deutsche Forschungsgemeinschaft (DFG, German Research Foundation) under Germany´s Excellence Strategy – EXC 2033 – 390677874 – RESOLV (S.K., M.H.). We used core facilities of the Center for Solvation Science ZEMOS funded by the German Federal Ministry of Education and Research BMBF and by the Ministry of Culture and Research of North Rhine-Westphalia. S.N acknowledges support by the Research Training Group "Confinement-controlled Chemistry" under Grant GRK2376-331085229 from the DFG. We thank the DFG and the VW foundation for funding our work on carbon nanotubes (S.K.). We acknowledge the support of the NSF CBET-2106587 award (L.V) and the computer time provided by the Texas Advanced Computing Center (TACC). We thank S. Ramos for her initial help with the THz experiments and for her helpful comments. We thank J. T. Metternich for providing the code used in supplementary Fig. 5. We thank B. König for providing the FTIR schematic used in Supplementary Fig. 13.

## Author contributions

Designed research: M.H. and S.K. Analyzed Data: S.N., P.G., L.S., and C.H. Funding acquisition: M.H., S.K., and L.V. Performed experimental research: S.N. P.G., L.S.; Performed simulation: P.K and L.V. Contributed analytical tools: C.H. Wrote manuscript: S.N. P.G., M.H., S.K., and L.V.; Review and editing: all authors.

## Funding

## Competing interests

The authors declare no competing interest.
