## [Peer Review File · Nature Communications]

THz coupling in carbon nanotube sensors modulates their fluorescenceREVIEWER COMMENTS

Reviewer #1 (Remarks to the Author):

This paper studies the mechanism through which the fluorescence of carbon nanotubes is modulated by small molecules in solution, through a combination of fluorescence and THz absorption spectroscopy, supported by molecular dynamics simulations. The authors propose that the water surrounding the carbon nanotubes provides an additional non-radiative decay pathway for the excitons, whose efficiency is modulated by the small molecules. To my knowledge, the de-excitation mechanisms of carbon nanotubes in solution have not been thoroughly established, and the idea of exciton coupling to the THz vibrations of water is new, and interesting not only for molecular sensing, but also in the context of nanoscale fluid transport. Nevertheless, I think that the proposed mechanism is not fully supported by the data, and definitively establishing it would require further experiments.

It is a priori clear that a change in the fluorescence quantum yield reflects a change in non-radiative decay pathways for the exciton. The authors show a correlation between the quantum yield and the THz absorption of the solution, but it is not clear that this correlation implies a causality. It could be, for instance, that the CNT environment affects the transition rate of the exciton to a dark state, and the de-excitation rate of this dark state (possibly controlled by the coupling to water) does not affect the quantum yield. I think that causality can be established only through a pump-probe experiment, if it shows a correlation between the rate of change in the THz signal after photoexcitation and the quantum yield. I am also confused by the conclusion that the increase in THz absorption upon addition of CNTs to the solution reflects the formation of water/CNT coupled modes. It could, for instance, simply reflect the THz absorption of the CNT/DNA system. How can the authors be sure that water is implicated in the THz modes that, according to them, provide the non-radiative decay pathway for the exciton?

Minor points:

- In the title, it is unclear what the word “coupling” refers to. What is coupled to what?
- The work from the Cox group is indeed simulations, but the work from the Bocquet group is analytical theory. There is now also experimental evidence for the coupling of THz modes at the solid-liquid interface (Nat. Nanotechnol. 18, 898–904 (2023)).

- I think that the paper could be improved by clarifying and shortening the section on simulations. “We employed bash and python scripts ...” is an overly detailed description of basic methods. To me, this is similar to “We employed a pipette to put the solution into the spectrometer”.
- Standard error rather than standard deviation should be used for the error bars.

Reviewer #2 (Remarks to the Author):

The work from Nalige et al. presents a THz study of the local hydration of solvated single-wall carbon nanotubes (SWCNTs) upon binding of the analytes dopamine and riboflavin. The authors couple experimental evidence and simulation to rationalise a linear (inverse) correlation between changes in THz amplitude and the intensity of the change in fluorescence induced by the analytes. In particular, the investigation suggests that THz amplitude signals can probe the charge fluctuations in the SWCNTs from the charge fluctuation in the hydration layer. In this study, ssDNA and surfactants such as deoxycholate, sodium cholate, and sodium dodecyl benzene sulphonate, were employed to disperse SWCNTs in aqueous solution. The authors found that the organic corona of the surfactants shapes the local water (which determines the exciton dynamics). Through this they conclude that THz signals can allow for quantitative predictions of signal transduction and for inferring principles to optimize fluorescent biosensors by design.

By and large, showing that THz screening could serve as a predictor for the fluorescence yield, hence allowing to better understand how binding of an analyte ultimately affects the optical properties of SWCNT-based fluorescent nano sensors, can allow to develop novel design principles for biosensors with stronger signal transduction. This is surely of interest to the broad readership of Nature Communications.

However, I am of the opinion that in its present form, the work submitted here would require major revisions prior publication. I have listed below some clarifications and potential amendments/additions that I invite the authors to consider in order to strengthen their manuscript to the level required for publication in Nature Communications.

First of all, many supporting information figures are not mentioned in the main manuscript

(e.g. figures S6 , S7 and S8) making this reader wonder if the data reported in those figures has been properly discussed (for example they contain some control experiments that in my opinion the authors do not discuss/highlight enough in in the main text).

In general ,the supporting information figures should be presented in sequential order, while here Figure S3 is mentioned after S4 in main text (page 8) , and figs S11 and S12 are mentioned before S9 and S10 .

The data obtained using (AT)15 is presented in figure 3 but not discussed in the main text for that figure , while (AT)15 data is only discussed in the main text once figure 4 is presented/discussed : please amend and add a discussion also when presenting the data in figure 3 (even figure 4 partly summarise it) .

Can the authors comment on the potential effect of changing the pH of the buffer or changing more systematically the ionic strength? Would it be worth considering some experiments in that regard ?

Would the authors consider performing experiments with serotonin , to compare with the results obtained with dopamine and show selectivity effects on the THz signals, also vs fluorescence?

Other minor points: the font of the text in all three images of figure 2 should be increased . More importantly, in the caption of figure 2 there is a mistake when saying “same as b” should be “same as a”

I also encourage the authors to expand on the last sentence, maybe reiterating the main point , when they write “The model presented here, gives now a conclusive explanation for the underlying molecular mechanism”.

In summary, I encourage the authors to consider the aforementioned points and revise their manuscript accordingly , as I believe the topic and issues addressed in their study can deserve publication in Nature Communications (after revision).

Reviewer #3 (Remarks to the Author):

The manuscript discusses the effect of coupling between DNA-single wall carbon nanotubes and hydration shell on their fluorescence and terahertz spectrum, which could be considered useful in design of SWCNT based biosensors. The results presented in this manuscript are valuable and interesting in a practical point of view and could be considered for publication, after following revision.

- The experimental setup of the utilized Terahertz spectroscopy measurement should be presented. It would help the readers to find out whether the SWCNTs are probed simultaneously with a low intensity laser pulse and in this case how the effects of the laser pulse has been included in the analyses. If a laser pulse has not been present through a THz-TDS setup, how such broadband THz emission has been provided?
- Excitation, absorption and optoelectronic properties of SWCNTs over visible and terahertz frequency ranges are determined by their structural features. The complex electric permittivity and therefore the dispersion properties of SWCNTs significantly depend not only on their chirality, which determines the diameter and rolling angle, but also on the filling factor and alignment of SWCNTs. Therefore, different results would be expected for SWCNTs of various structural conditions. For the SWCNT chirality employed in the present work it would be also difficult to exclude the influence of other structural initial conditions, those which have not been probably in control, on the obtained results. These structural conditions should be properly mentioned and declared in the manuscript by referring to those discussed in the literature such as: Optics Express Vol. 29, 38359 (2021), <https://doi.org/10.1364/OE.442168>.

Point-by-Point Response

Manuscript NCOMMS-23-56031

We thank all reviewers for the careful evaluation of our manuscript. All questions and points raised by the reviewers are addressed in the following point-by-point response with the discussion of additional data, figures, and references. The revised manuscript and supplementary information with marked changes are provided separately for revision.

Reviewer Comments

Reviewer #1 (Remarks to the author):

“This paper studies the mechanism through which the fluorescence of carbon nanotubes is modulated by small molecules in solution, through a combination of fluorescence and THz absorption spectroscopy, supported by molecular dynamics simulations. The authors propose that the water surrounding the carbon nanotubes provides an additional non-radiative decay pathway for the excitons, whose efficiency is modulated by the small molecules. To my knowledge, the de-excitation mechanisms of carbon nanotubes in solution have not been thoroughly established, and the idea of exciton coupling to the THz vibrations of water is new, and interesting not only for molecular sensing, but also in the context of nanoscale fluid transport. Nevertheless, I think that the proposed mechanism is not fully supported by the data, and definitively establishing it would require further experiments.”

Response:

We thank the reviewer for the overall positive evaluation. Following the advice of the referee we now provide additional data which support our proposed novel mechanism. These are described in detail below.

- a. “It is a priori clear that a change in the fluorescence quantum yield reflects a change in non-radiative decay pathways for the exciton. The authors show a correlation between the quantum yield and the THz absorption of the solution, but it is not clear that this correlation implies a causality. It could be, for instance, that the CNT environment affects the transition rate of the exciton to a dark state, and the de-excitation rate of this dark state (possibly controlled by the coupling to water) does not affect the quantum yield. I think that causality can be established only through a pump-probe experiment, if it shows a correlation between the rate of change in the THz signal after photoexcitation and the quantum yield.

Response:

Following the suggestion of the referee we have carried out additional ON - OFF experiments using a green LED (placed in the sample compartment) as a radiation source to optically excite the SWCNTs (at the E_{22} transition at 560 nm like all fluorescence spectroscopy experiments) within the FTIR spectrometer. We tested DOC-SWCNTs and DNA-SWCNTs. Fig. R1 (Fig. 2b in the main manuscript) shows the fluorescence intensities at the same concentration/absorption and therefore corresponds to the quantum yield. DOC-SWCNTs have a higher quantum yield than DNA-SWCNTs. In Fig. R2 we plot the difference in THz absorption between light on (excitons are created) and LED light off (no excitons). In both cases, the presence of electronic excitation or more specifically excitons in the SWCNTs affects the THz response ($\Delta\alpha$). If the THz signal, reporting on the charge fluctuations in water, would not be affected, we would see a flat spectral line around zero. Interestingly, we find that $\Delta\alpha$ upon excitation is smaller for DOC-SWCNTs than for DNA-SWCNTs, whereas for the quantum yields the opposite holds. In summary, this data supports our hypothesis, that energy transfer via fluorescence and exciton–hydration coupling are two alternative pathways in this system.

Figure R1/Figure 2b: Fluorescence spectra of SWCNTs with respective polymers.

Figure R2: $\Delta\alpha$ ($= \alpha_{\text{LED ON}} - \alpha_{\text{LED OFF}}$) of DNA-SWCNTs and DOC-SWCNTs.

- b. “I am also confused by the conclusion that the increase in THz absorption upon the addition of CNTs to the solution reflects the formation of water/CNT coupled modes. It could, for instance, simply reflect the THz absorption of the CNT/DNA system. How can the authors be sure that water is implicated in the THz modes that, according to them, provide the non-radiative decay pathway for the exciton?”

Response:

In a series of joint experimental/simulations studies we have demonstrated that in the frequency region between 50 and 450 cm^{-1} we cannot separate solute and solvent modes, instead, we observe correlated solute solvent modes extending even over several hydration shells. For glycine, we could show that the observed broad THz spectrum contain distinct modes stemming from translational motions of the glycine with localized water molecules (around 100 cm^{-1}), around 200 cm^{-1} we observe a correlated glycine water network mode reporting on the coupling between the zwitterionic glycine and water modes. Above 400 cm^{-1} we observe an intramolecular mode, which is also identified by a smaller linewidth (JACS 2014, 136, 5031, Havenith et al.).

In a similar way, for ions, we identified ion specific so-called rattling modes, i.e. a motion of the charge within its hydration cage. In later study we disclose that theoretical THz difference spectra of aqueous salt solutions can be deciphered in terms of only a handful of dipolar auto- and crosscorrelations, including the second solvation shell. WE concluded that very distinct intermolecular (auto- and cross-) correlations shape the total THz responses of anions and cations in water on the basis of a subtle intensity compensation mechanism. An in summary: „These molecular underpinnings of the THz line shape of aqueous salt solutions cannot be unveiled by directly subtracting the spectra of either the solutions or of the individual ions from the pure bulk water reference spectrum—neither experimentally nor computationally.“ The same holds in the case presented here in this paper:

We have also measured dry SWCNTs and DNA-SWCNTs with an ATR (attenuated total reflection) FTIR unit (because the sample is solid and cannot directly be measured in the standard transmission cell with two diamond windows). The absorption spectrum of solid SWCNTs (Figure R3, now also included in SI as S16) differs considerably from the spectrum of the solvated DNA-SWCNT system in our transmission measurements (Fig. 2a in the main manuscript). We observe a characteristic increase in absorption below 1 THz (33 cm^{-1}), which is also seen for graphene. (<http://dx.doi.org/10.1063/1.4944531>; <https://doi.org/10.1016/j.carbon.2018.02.005>) The spectral response that we see is a correlated SWCNT / water spectrum. Hence, we assign the broadband absorption observed (e.g. Fig. 2a in the main manuscript) to the coupling of electronic motions in SWCNTs (plasmons) to charge fluctuations of water and not to the polymer-SWCNT itself.

We want to add that we also measured bare ssDNA and ssDNA + Analytes in PBS buffer. These do not show any significant absorption in the THz range (Fig. R4).

DNA-SWCNT system responds in different ways to the different analytes (riboflavin, dopamine) in fluorescence and THz spectra, thus, these cannot be attributed to the absorption DNA-SWCNT only.

Figure R3: THz absorption of solid pristine SWCNT and DNA-SWCNT THz spectra measured on a diamond crystal of the ATR setup.

Figure R4: THz difference spectrum, $\Delta\alpha$ ($= \alpha_{\text{sample}} - \alpha_{\text{water}}$) of DNA and DNA + analyte measured in the transmission cell and with the same conditions as the DNA-SWCNT system.

c. “In the title, it is unclear what the word “coupling” refers to. What is coupled to what? “

Response:

We thank the referee for this hint. In the studied frequency range, the observed modes in the spectra extend over several Angstroms, thus these are collective modes. This implies that we cannot assign these either to just water or solute (DNA-SWCNT) modes. For glycine, we have observed glycine-water hydrogen bond stretching modes at about 200 cm^{-1} . See Sun, Niehues, Forbert, Decka, Schwaab, Marx, Havenith, JACS 136, 5031 and Angew. Chem. Int. Ed. 2021, 60,3768–3772. For ions, the low-frequency spectrum could be deciphered in terms of only a handful of dipolar auto- and cross-correlations, including the second solvation shell. This emphasizes the importance of cross-correlations being often neglected in multicomponent models. (Schienbein. P., et al DOI: 10.1021/acs.jpcllett.7b00713).

Based on this we have added the below discussion on page 9 of the manuscript:

“For the SWCNTs, it has been proposed that there is a strong coupling between surface plasmon modes in SWCNTs and excitons (<http://dx.doi.org/10.1103/PhysRevB.80.085407>). As the energy of (axial) surface plasmons of SWCNTs is in the THz range (<https://doi.org/10.1016/j.carbon.2020.05.019>) we propose that there is a dissipation pathway via the coupling of excitons and THz modes of water are coupled. An alternative could be the large number of (dark) states within the bandgap of the SWCNT.“

- d. “The work from the Cox group is indeed simulations, but the work from the Bocquet group is analytical theory. There is now also experimental evidence for the coupling of THz modes at the solid-liquid interface (Nat. Nanotechnol. 18, 898–904 (2023)).”

Response:

We had indeed cited two of these papers and now also cite the third one and clarify the text in the manuscript on page 4. We thank the referee for pointing this out.

- e. “I think that the paper could be improved by clarifying and shortening the section on simulations. “We employed bash and python scripts ...” is an overly detailed description of basic methods. To me, this is similar to “We employed a pipette to put the solution into the spectrometer”.

Response:

Following the advice of the referee we have transferred descriptions of simulation methods to the supplementary information and changed the text accordingly on pages 11 and 12 of the manuscript:

“Given these results of the bulk analysis, the localized contact areas between the SWCNT surface segments and surrounding water molecules were examined next (Fig. S11).”

- f. “Standard error rather than standard deviation should be used for the error bars.”

Response:

We have changed the error bars from standard deviation to standard error and changed the captions accordingly.

Reviewer #2 (Remarks to the author):

“The work from Nalige et al. presents a THz study of the the local hydration of solvated single-wall carbon nanotubes (SWCNTs) upon binding of the analytes dopamine and riboflavin. The authors couple experimental evidence and simulation to rationalise a linear (inverse) correlation between changes in THz amplitude and the intensity of the change in fluorescence induced by the analytes. In particular, the investigation suggest that THz amplitude signals can probe the charge fluctuations in the SWCNTs from the charge fluctuation in the hydration layer. In this study, ssDNA and surfactants such as deoxycholate, sodium cholate, and sodium dodecyl benzene sulphonate, were employed to disperse SWCNTs in aqueous solution. The authors found that the organic corona of the surfactants shapes the local water (which determines the exciton dynamics). Through this they conclude that THz signals can allow for quantitative predictions of signal transduction and for inferring principles to optimize fluorescent biosensors by design. By and large, showing that THz screening could serve as a predictor for the fluorescence yield, hence allowing to better understand how binding of an analyte ultimately affects the optical properties of SWCNT-based fluorescent nano sensors, can allow to develop novel design principles for biosensors with stronger signal transduction. This is surely of interest to the broad readership of Nature Communications.”

Response:

We thank the reviewer for the overall positive evaluation.

- a. “However, I am of the opinion that in its present form, the work submitted here would require major revisions prior publication. I have listed below some clarifications and potential amendments/additions that I invite the authors to consider in order to strengthen their manuscript to the level required for publication in Nature Communications. First of all, many supporting information figures are not mentioned in the main manuscript (e.g. figures S6 , S7 and S8) making this reader wonder if the data reported in those figures has been properly discussed (for example they contain some control experiments that in my opinion the authors do not discuss/highlight enough in in the main text). In general, the supporting information figures should be presented in sequential order, while here Figure S3 is mentioned after S4 in main text (page 8), and figs S11 and S12 are mentioned before S9 and S10 .”

Response:

We are sorry for not explicitly discussing all figures from the SI and thank the referee for pointing this out. We have ordered the SI figures to match the storyline in the main manuscript and mentioned all SI figures in the main text. Figure S8, which shows the initial configurations of the systems modeled in MD simulations and the intermediate results of the computational protocol we followed, is discussed in section 3 of the Supporting Information document. This section describes the details of the computational methodology. Even though Figure S8 is not discussed in the main text, it provides computational details essential for the reproducibility of the reported modeling.

- b. “The data obtained using (AT)₁₅ is presented in figure 3 but not discussed in the main text for that figure, while (AT)₁₅ data is only discussed in the main text once figure 4 is presented/discussed: please amend and add a discussion also when presenting the data in figure 3 (even figure 4 partly summarise it)”

Response:

We have added the description of the (AT)₁₅ dataset on page 8 of the main manuscript

“Additionally, we carried out the above-mentioned experiments with (AT)₁₅-SWCNTs (Fig. 3b, d, S4). The fluorescence and THz absorption responses of (AT)₁₅-SWCNTs followed the same trend as seen for (GT)₁₀-SWCNTs but the magnitude of change in fluorescence upon addition of analytes is smaller. Since the coupling of the electronic fluctuations of SWCNT to charge density fluctuations in water depends on the specific SWCNT corona (Fig. 2), the alterations induced by the analytes can vary in magnitude.”

Furthermore, we summarised the main message of Figures 3 and 4 as recommended on page 10 of the manuscript:

“Hence, the SWCNT corona is impacted by the analyte. In general, analytes that increase fluorescence, such as dopamine, reduce the observed THz coupling efficiency, whereas quenching analytes, such as riboflavin, increase the coupling efficiency.”

- c. “Can the authors comment on the potential effect of changing the pH of the buffer or changing more systematically the ionic strength? Would it be worth considering some experiments in that regard?”

Response:

The analytes considered in this study do not change the pH value or ionic strength under our experimental conditions in the buffered (PBS) system. pH and ion strength are known to affect the fluorescence of SWCNTs (see e.g. Zheng et al. 2020, DOI: 10.1021/acsnano.0c05720 and Gillen et al. 2019, DOI: 10.1021/acsnano.0c05720). Therefore, we expect the THz signals to change as well as the fluorescence changes. We did not perform experiments in this direction so far because we wanted to understand first how sensing works under physiological conditions. In the future, one could extend this to other conditions as suggested by the referee..

- d. “Would the authors consider performing experiments with serotonin, to compare with the results obtained with dopamine and show selectivity effects on the THz signals, also vs fluorescence?”

Response:

Thank you for this suggestion. The distinction between monoamine neurotransmitters is a challenge for the whole field of analytical chemistry because of their similarity. For example, electrochemical methods have worked on this challenge for decades and it is still not solved. DNA-SWCNTs (e.g. (GT)₁₀) differ in their response to the catecholamine neurotransmitters dopamine, epinephrine, and epinephrine (Mann *et al.* 2017, DOI: 10.3390/s17071521). To our knowledge serotonin does not cause strong fluorescence changes of (GT)₁₀ – SWCNTs and we had therefore used aptamers to create a serotonin sensor (see Dinarvand *et al.* Nano Letters 2019, DOI: 10.1021/acs.nanolett.9b02865). However, we have carried out additional measurements for the dopamine homolog epinephrine. In line with our hypothesis, we observed a decrease in $\Delta\alpha$ of THz measurements (Fig. R5), due to the reduced coupling and in turn an increase in fluorescence (Fig. R6). We observe a decrease in the $\Delta\alpha$ of the THz measurements for both dopamine and epinephrine. The changes in fluorescence (Figure 3c, d and Figure R5/R6) amount to an increase of a 50 % (epinephrine) versus 125 % (dopamine). Figures R5 and R6 have been added to SI figures, S1 and S2. These differences in selectivity for very similar molecules are a highly interesting topic that will be further explored in the future.

Figure R5: THz difference spectra, $\Delta\alpha$ ($= \alpha_{\text{sample}} - \alpha_{\text{water}}$) of (GT)₁₀-SWCNT without and with the analyte epinephrine.

Figure R6: Fluorescence response of (GT)₁₀-SWCNT upon addition of epinephrine. a. Fluorescence spectra before and after the addition of 1 % (V/V) PBS buffer as control and 100 μM epinephrine (end concentration). b. Relative fluorescence change. Error bars are standard errors of triplicates.

- e. “Other minor points: the font of the text in all three images of Figure 2 should be increased. More importantly, in the caption of Figure 2 there is a mistake when saying “same as b” should be “same as a”

Response

We thank the referee for pointing this out. We increased the font sizes in all figures equally and corrected our mistake.

- f. “I also encourage the authors to expand on the last sentence, maybe reiterating the main point, when they write “The model presented here, gives now a conclusive explanation for the underlying molecular mechanism”. In summary, I encourage the authors to consider the aforementioned points and revise their manuscript accordingly, as I believe the topic and issues addressed in their study deserve publication in Nature Communications (after revision).”

Response:

As per the suggestion of the referee we have expanded the conclusion in the manuscript on page 13 as follows:

“Here, we investigate the coupling of the low-frequency DNA-SWCNT mode with the hydration water via THz absorption. We propose that the change in absorption reports on the energy transfer between exciton-coupled surface plasmons of DNA-SWCNTs and the translational and hydrogen bond stretch mode. The efficiency of this coupling is proposed to affect the magnitude of the broadband correlated solute-water mode, and can be modified by the binding of analytes –depending on the binding motif. Upon optical excitation, the energy can either be transferred efficiently to the solvent (via correlated DNA- SWCNT/ hydration water modes), which are proposed to be similar in the ground and excited electronic state, or will finally lead to an exciton decay via fluorescence. Thus, in this picture excitons undergo radiative decay (fluorescence) or non-radiative decay (e.g., quenching at a SWCNT end or transition to a dark state). While it was known before, that these transition rates are affected by the presence of the analyte, the physicochemical origin remained unknown. Now, we show that

coupling between the DNA-SWCNTs and the hydration shell dictates these transition rates. The exciton screens the hydration shells, which are affected by analyte binding³⁴. The pathway for energy release of an exciton is twofold: either energy is released via mode coupling into the solvent or via fluorescence. Therefore, THz screening serves as a predictor for the fluorescence yield of SWCNT-based fluorescent nanosensors.”

Reviewer #3 (Remark to the author):

The manuscript discusses the effect of coupling between DNA-single wall carbon nanotubes and hydration shell on their fluorescence and terahertz spectrum, which could be considered useful in design of SWCNT based biosensors. The results presented in this manuscript are valuable and interesting in a practical point of view and could be considered for publication, after following revision.

Response:

We thank the reviewer for the positive evaluation of our work.

- a. "The experimental setup of the utilized Terahertz spectroscopy measurement should be presented. It would help the readers to find out whether the SWCNTs are probes simultaneously with a low intensity laser pulse and in this case how the effects of the laser pulse has been included in the analyses. If a laser pulse has not been present through a THz-TDS setup, how such broadband THz emission has been provided?"

Response:

We used a mercury lamp as a radiation source (emission from 20 cm^{-1} up to 500 cm^{-1}). Mylar multilayer served as a beam splitter, thereby restricting our frequency range from 50 cm^{-1} up to 400 cm^{-1} . Following the advice of the referee we provided a description and figure of the experimental setup in the SI (Figure S13).

Figure R7/Figure S13. Schematic of the FTIR setup. The sample compartment is temperature stabilized and purged with dry nitrogen to ensure stable operating conditions during measurements.

- b. “Excitation, absorption and optoelectronic properties of SWCNTs over visible and terahertz frequency ranges are determined by their structural features. The complex electric permittivity and therefore the dispersion properties of SWCNTs significantly depend not only on their chirality, which determines the diameter and rolling angle, but also on the filling factor and alignment of SWCNTs. Therefore, different results would be expected for SWCNTs of various structural conditions. For the SWCNT chirality employed in the present work it would be also difficult to exclude the influence of other structural initial conditions, those which have not been probably in control, on the obtained results. These structural conditions should be properly mentioned and declared in the manuscript by referring to those discussed in the literature such as: Optics Express Vol. 29, 38359 (2021), <https://doi.org/10.1364/OE.442168>.”

Response:

We thank the referee for raising this point. We have now included a discussion of the influence of SWCNT chirality on sensing and added the suggested citation.

All experiments were conducted with samples using (6,5)-enriched SWCNTs produced via the CoMoCAT process and purchased from Merck. It contains > 40 % (6,5) chirality and ≥ 95 % semiconducting SWCNTs, with an average diameter of 0.78 nm (i.e., the diameter of (6,5)-SWCNTs). All samples were functionalized in a buffer following the same protocol to ensure comparability. All experiments were also performed in solution and therefore there is no preferential orientation of the SWCNTs that could have affected fluorescence and THz measurements.

What indeed could affect the results is the structure of the SWCNTs (chirality (n,m)). It is known that for DNA-SWCNT-based sensors for dopamine or riboflavin, chirality does not play a strong role (Nißler et. al. 2021, DOI: 10.1021/acs.analchem.1c00168).

However, this does not rule out different responses with different surface chemistry as observed for other surface chemistries (Kim et al. Nature Biomedical Engineering 2022).

We, therefore, analyzed the fluorescence responses for three chiralities [(6,4), (6,5), (9,4)] in the NIR spectrum of CoMoCAT SWCNTs. The results are qualitatively similar (Fig. R8- also added to SI as SI5) although there are a few differences. A quantitative comparison of the change in fluorescence is difficult due to the differences in concentration of the individual chiralities. For previous monochiral experiments, we showed (as discussed above) that the influence of chirality on sensing is minor compared to the surface chemistry (e.g. DNA sequence).

Figure R8/Figure S5: Relative change of fluorescence intensity of distinct SWCNT chirality upon the addition of 100 μ M of the corresponding analyte. Error bars are standard errors of triplicates.

We added the following sentences to clarify the discussed aspects and add the mentioned citation on page 8 of the manuscript: “The chirality of the SWCNTs could play a certain role in the sensing mechanism. In our experiments the main species were (6,5)-SWCNTs. It was shown before that the chirality of DNA-SWCNT based sensors for small molecules play a less important role than the surface chemistry (e.g. DNA sequence³⁸, compare Fig. S5), Other factors such as orientation could also play a role but should be negligible in solution³⁹”.

We thank all referees for the helpful comments that improved the manuscript and we hope it can now be accepted for publication in Nature Communications.

REVIEWERS' COMMENTS

Reviewer #1 (Remarks to the Author):

I think that the authors have provided convincing justification for the points that were questionable in the original manuscript. The additional explanations and data demonstrate indeed that the modulation of the THz fluctuations in the nanotube's hydration shell are likely the cause of changes in fluorescent quantum yield. I am overall satisfied with the revision.

I am still somewhat confused by the title: the word "coupling" does not make sense to me if it is not stated which two entities are coupled. I would suggest, perhaps, "Coupling to THz fluctuations of solvent environment in carbon nanotube sensors modulates their fluorescence".

Reviewer #2 (Remarks to the Author):

The authors addressed the points I highlighted in my initial review evaluation . Therefore I recommend publication of this work in Nature Communication

Reviewer #3 (Remarks to the Author):

The reviewers' comments have been adequately addressed and the revised version of the manuscript has been well improved. In my opinion it could be considered for publication.